# Genome wide analysis reveals heparan sulfate epimerase modulates TDP-43 proteinopathy

Nicole F. Liachko[1,2�], Aleen D. Saxton[1☰], Pamela J. McMillan[3], Timothy J. Strovas[1], C. Dirk Keene[4], Thomas D. Bird[1,5,6], Brian C. Kraemer[1,2,3,6]*

**1** Geriatrics Research Education and Clinical Center, Veterans Affairs Puget Sound Health Care System, Seattle, Washington, United States of America, **2** Division of Gerontology and Geriatric Medicine, Department of Medicine, University of Washington, Seattle, Washington, United States of America, **3** Department of Psychiatry and Behavioral Sciences, University of Washington, Seattle, Washington, United States of America, **4** Department of Pathology, University of Washington, Seattle, Washington, United States of America, **5** Department of Neurology, University of Washington, Seattle, Washington, United States of America, **6** Division of Medical Genetics, Department of Medicine, University of Washington, Seattle, Washington, United States of America

☰ These authors contributed equally to this work.
* kraemerb@u.washington.edu

**Data Availability Statement:** All relevant data are within the manuscript and its Supporting Information files.

## Abstract

Pathological phosphorylated TDP-43 protein (pTDP) deposition drives neurodegeneration in amyotrophic lateral sclerosis (ALS) and frontotemporal lobar degeneration (FTLD-TDP). However, the cellular and genetic mechanisms at work in pathological TDP-43 toxicity are not fully elucidated. To identify genetic modifiers of TDP-43 neurotoxicity, we utilized a *Caenorhabditis elegans* model of TDP-43 proteinopathy expressing human mutant TDP-43 pan-neuronally (TDP-43 tg). In TDP-43 tg *C. elegans*, we conducted a genome-wide RNAi screen covering 16,767 *C. elegans* genes for loss of function genetic suppressors of TDP-43-driven motor dysfunction. We identified 46 candidate genes that when knocked down partially ameliorate TDP-43 related phenotypes; 24 of these candidate genes have conserved homologs in the human genome. To rigorously validate the RNAi findings, we crossed the TDP-43 transgene into the background of homozygous strong genetic loss of function mutations. We have confirmed 9 of the 24 candidate genes significantly modulate TDP-43 transgenic phenotypes. Among the validated genes we focused on, one of the most consistent genetic modifier genes protecting against pTDP accumulation and motor deficits was the heparan sulfate-modifying enzyme *hse-5*, the *C. elegans* homolog of glucuronic acid epimerase (*GLCE*). We found that knockdown of human *GLCE* in cultured human cells protects against oxidative stress induced pTDP accumulation. Furthermore, expression of glucuronic acid epimerase is significantly decreased in the brains of FTLD-TDP cases relative to normal controls, demonstrating the potential disease relevance of the candidate genes identified. Taken together these findings nominate glucuronic acid epimerase as a novel candidate therapeutic target for TDP-43 proteinopathies including ALS and FTLD-TDP.

**Funding:** We thank the National Institutes of Health R01NS064131 (BCK) and P50AG005136 (CDK– UW ADRC Neuropathology Core). We also thank the Department of Veterans Affairs (Merit Review Grant I01BX003755 to BCK and Merit Review Grant I01BX004044 to NFL) for funding and training support. We also thank the Nancy and Buster Alvord endowment (CDK). The funders had no role in study design, data collection and analysis, decision to publish, or preparation of the manuscript.

**Competing interests:** The authors have declared that no competing interests exist.

## Author summary

The protein TDP-43 forms aggregates in disease-affected neurons in patients with ALS and FTLD-TDP. In addition, mutations in the human gene coding for TDP-43 can cause inherited ALS. By expressing human mutant TDP-43 protein in *C. elegans* neurons, we have modelled aspects of ALS pathobiology. This animal model exhibits severe motor dysfunction, progressive neurodegeneration, and accumulation of abnormally modified TDP-43 protein. To identify genes controlling TDP-43 neurotoxicity in *C. elegans*, we have conducted a genome-wide reverse genetic screen and found 46 genes that participate in TDP-43 neurotoxicity. We demonstrated that one of them, glucuronic acid epimerase, is decreased in patients with FTLD-TDP suggesting inhibitors of glucuronic acid epimerase could have therapeutic value for ALS and FTLD.

## Introduction

Amyotrophic lateral sclerosis (ALS) is a devastating neurodegenerative disease characterized by loss of motor neurons, leading to progressive muscle wasting and death typically by respiratory failure. Aggregates of the protein TDP-43 in disease-affected neurons are the hallmark of most cases of ALS, while mutations in the gene coding for TDP-43, *TARDBP*, have been identified in ~4% of families with heritable ALS directly linking TDP-43 dysfunction with disease [1]. TDP-43 is also the primary aggregating protein in about 50% of cases of frontotemporal lobar degeneration (FTLD-TDP) [2]. Furthermore, TDP-43 positive aggregates occur as secondary pathologies in a subset of other neurodegenerative diseases including Alzheimer's disease (AD) [3]. The presence of TDP-43 co-pathology correlates with a greater likelihood of clinical dementia and more rapid cognitive decline after diagnosis, indicating the presence or absence of pathological TDP-43 influences disease severity [4, 5]. Given the prevalence of TDP-43 pathology in neurodegenerative diseases, it is critical to understand the molecular pathways that interact with TDP-43 and promote disease.

Within inclusions in disease-affected neurons, TDP-43 exhibits post-translational modifications not observed in healthy neurons, including shorter C-terminal protein species, ubiquitination, acetylation, SUMOylation, and phosphorylation [6–9]. Of these modifications, abnormal phosphorylation of TDP-43 represents the most consistent pathological feature of ALS and FTLD-TDP; pTDP is used as a neuropathological marker to identify TDP-43-positive protein inclusions in brain and spinal cord post-mortem [10]. Phosphorylation at S409/410 of TDP-43 (pTDP) potentiates a number of toxic disruptions in normal TDP-43 metabolism, including decreased TDP-43 protein turnover, increased TDP-43 stabilization, cellular mislocalization of TDP-43, protein aggregation, and neurodegeneration [11–16]. Phosphorylation may also alter the ability of TDP-43 to appropriately participate in liquid-liquid phase separated membrane-less organelles or granules [17, 18].

Unbiased molecular genetic screens in a variety of model systems provide a powerful way to identify novel modifiers of proteinopathy driven neurodegeneration and to elucidate key cellular pathways that contribute to disease [19–26]. Several reverse genetic screens targeting pathological TDP-43 have been performed in model systems including yeast, *Drosophila*, and cultured cells [27–33], and have nominated TDP-43 modifiers including ataxin 2, Dbr1, ITPR1, Wnd, p38, JNK, GSK, hat-trick, xmas-2, hnRNPs, and TCERG1, among others. These modifiers implicate diverse pathways affecting disease, including polyglutamine repeat size, RNA lariat debranching, mRNA splicing and export, nucleocytoplasmic shuttling, innate immunity, chromatin remodeling, and TDP-43 autoregulation. These screens differ in their

methods of modeling TDP-43 toxicity as well as the readouts for suppression or enhancement of phenotypes. Moreover, the hits identified among them are also predominantly non-overlapping indicating these screens have not saturated the identification of TDP-43 modifiers. It is likely that additional screening may identify new modifiers with relevance to TDP-43 mechanisms of disease.

To model TDP-43 proteinopathy, we previously developed a *Caenorhabditis elegans* model that expresses full-length familial ALS (fALS) mutant TDP-43 pan-neuronally [16]. *C. elegans* provides a tractable, simple model useful for genetic manipulations, behavioral assays, biochemistry, *in vivo* imaging, and large-scale screening. Importantly, they also have a well-characterized differentiated nervous system that includes the major neuronal types found in humans [34]. TDP-43 transgenic *C. elegans* (TDP-43 tg) exhibit phosphorylation of TDP-43 at S409/410, accumulation of insoluble TDP-43, severe motor abnormalities, neurodegeneration, and shortened lifespan [16]. In *C. elegans*, prevention of phosphorylation at S409/410 protects against TDP-43 neurotoxic phenotypes.

Using this *C. elegans* model of ALS, we have conducted a genome-wide RNAi screen for genes that control TDP-43-driven phenotypes. This is the first large-scale genetic screen in *C. elegans* for TDP-43 modifying genes. From this work, we have identified genetic suppressors of TDP-43 neurotoxicity that fall into a variety of molecular categories. Finally, we have shown that the extracellular matrix modifying enzyme HSE-5/ GLCE impacts TDP-43 pathology and is decreased in FTLD-TDP.

## Results

To identify genes and molecular pathways that modify TDP-43 toxicity, we performed an unbiased genome-wide RNAi screen in a *C. elegans* model of TDP-43 proteinopathy. This model expresses human fALS mutant TDP-43 (M337V) pan-neuronally (TDP-43 tg) and exhibits robust progressive motor dysfunction [16]. We used this phenotype to screen for visible suppression of motor dysfunction following RNAi-mediated gene inactivation. 16,757 RNAi clones, targeting 86% of the *C. elegans* genome [35], were individually screened for changes in behavior relative to control treated animals (Fig 1A). Following first-pass screening for candidate suppressors of TDP-43 tg motor dysfunction, hits were retested in non-transgenic *C. elegans* to eliminate those that affect *C. elegans* movement independent of TDP-43. Candidates with no effects on non-transgenic *C. elegans* were then retested against TDP-43 tg *C. elegans* in three biological replicate experiments to confirm the initial screen results. After the retesting of first-pass hits, 46 suppressor RNAi clones were identified that consistently improved motor function (Table 1 and Supplemental Table 1). While all the RNAi targets identified are potentially interesting TDP-43 modifiers, we prioritized candidates with translational relevance to human disease for follow-up experiments. To this end, we surveyed for conserved human homologs of candidate genes using the NCBI basic local alignment search tool (BLAST) [36]. Of the TDP-43 tg modifying genes, we found 24 suppressors with significant homology to human genes (Table 1). These gene lists were then annotated for roles *in vivo* using both searches of published literature and predictive gene functions from protein domain architecture. These genes fall into several functional classes, including genes involved in energy production and metabolism (*cox-10*, F23F12.3, F55G1.5, *paqr-1*, and *cox-6A*), extracellular matrix and cytoskeleton (*col-89*, *gly-8*, *hse-5*, *sax-2*, *vab-9*, and *zig-3*), ion transport (*cnnm-3*, C13B4.1, *unc-77*), nucleic acid function (*dna-2*, *tbx-11*, *umps-1*), proteostasis (C47E12.3, *gpx-7*, *pcp-5*), and signaling (F31E3.2, F40B5.2, Y44E3A.4) (Fig 1B). As RNAi-mediated inactivation of these genes partially suppressed toxic TDP-43 phenotypes, their normal cellular

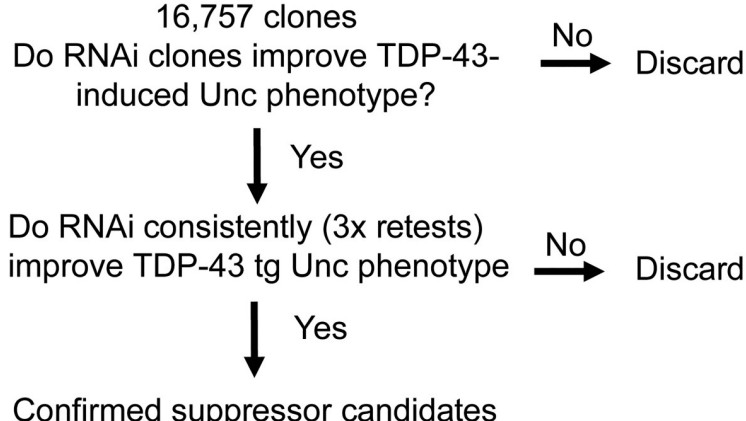

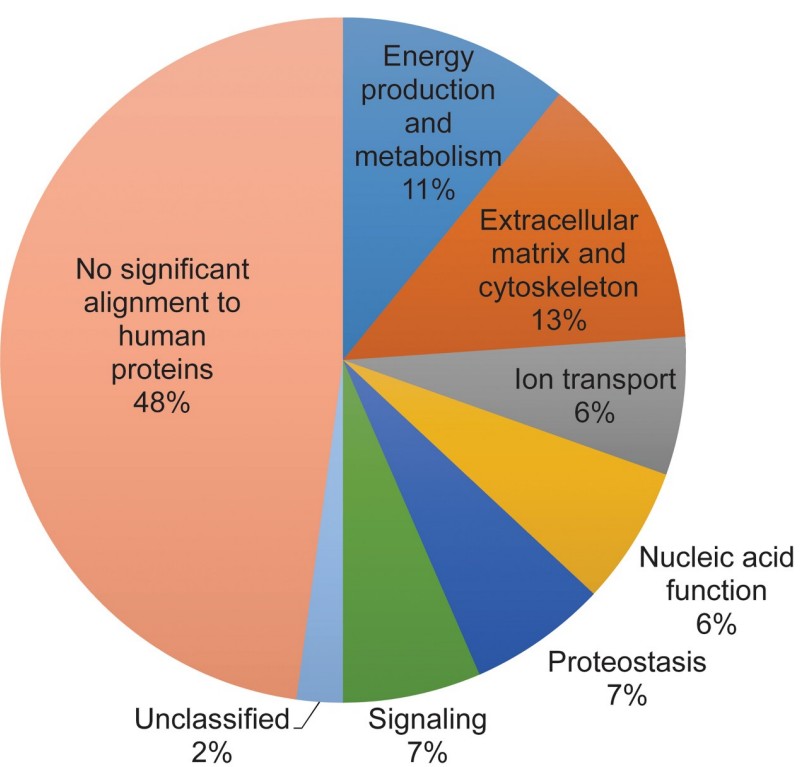

**Fig 1. Genome-wide RNAi screen identifies TDP-43 suppressors in several functional categories.** (*A*) Flowchart of the RNAi screen for modifiers of TDP-43-induced uncoordinated locomotion (Unc). (*B*) Percentage of suppressors identified in each functional group. Genes with no significant alignment within the human genome were not classified into functional groups.

functions may promote TDP-43 toxicity, making them potential targets for treating TDP-43 proteinopathy.

RNAi is a rapid and powerful tool to decrease gene expression *in vivo*. However, a single RNAi clone has the potential to target multiple genes simultaneously if the genes have a

**Table 1. TDP-43 modifying genes with human homologs.**

| C. elegans gene | HUGO symbol | BlastP e-value | Protein family/ domains |
|---|---|---|---|
| **Energy production and metabolism** | | | |
| cox-10 | COX10 | 9E-78 | Heme A:farnesyltransferase |
| F23F12.3 | SLC22A5 | 7E-36 | Organic cation/ carnitine transporter |
| F55G1.5 | SLC25A18 | 9E-68 | Mitochondrial glutamate/H+ symporter |
| paqr-1 | ADIPOR1/2 | 2E-116 | Adiponectin receptor |
| cox-6A | COX6A1 | 2E-14 | Cytochrome C oxidase (COX) structural component |
| **Extracellular matrix and cytoskeleton** | | | |
| col-89 | COL22A1 | 1E-12 | Fibrillar-associated collagen with interrupted triple helices |
| gly-8 | GALNT11 | 1E-101 | N-Acetylgalactosaminyltransferase |
| hse-5 | GLCE | 9E-113 | glucuronic acid epimerase |
| sax-2 | FRYL | 0E+00 | FRY protein |
| vab-9 | TMEM47 | 2E-12 | PMP22/EMP/claudin family |
| zig-3 | HMCN1 | 1E-09 | Immunoglobulin superfamily |
| **Ion transport** | | | |
| cnnm-3 | CNNM4 | 5E-118 | Ancient conserved domain |
| C13B4.1 | FRRS1 | 9E-23 | Cytochrome b561 family; ferric chelate reductase |
| unc-77 | NALCN | 0E+00 | Voltage-gated sodium channel |
| **Nucleic acid function** | | | |
| dna-2 | DNA2 | 2E-129 | DNA2/NAM7 helicase |
| tbx-11 | TBR1 | 2E-28 | T-box transcription factor |
| umps-1 | UMPS | 1E-100 | Uridine monophosphate synthetase |
| **Proteostasis** | | | |
| C47E12.3 | EDEM1 | 0E+00 | ER degradation-enhancing alpha-mannosidase |
| gpx-7 | GPX4 | 1E-39 | Glutathione peroxidase |
| pcp-5 | PRCP | 1E-114 | Prolylcarboxypeptidase |
| **Signaling** | | | |
| F31E3.2 | SGK494 | 5E-59 | Serine/ threonine kinase |
| F40B5.2 | TMPPE | 1E-67 | Metallophosphoesterase |
| Y44E3A.4 | SH3KBP1 | 1E-31 | SH3 domain |
| **Unclassified** | | | |
| T24H10.4 | TMEM53 | 7E-13 | DUF829 domain-containing |

stretch of identical nucleotide sequence. Thus, we used genetic loss-of-function mutations in candidate genes to confirm RNAi phenotypes. 15 existing *C. elegans* null mutations were available for candidate modifiers with human homologs (Table 2). We also selected two null mutations for testing from the candidates without human homologs. This cohort of mutants includes alleles from at least one gene per functional category, allowing us to test the contribution of these gene classes to TDP-43 phenotypes. Each of these mutant strains were crossed to TDP-43 tg *C. elegans* to evaluate changes in motor dysfunction. For genes that were present on the same chromosome as our primary TDP-43 tg transgenic strain (chromosome IV), we crossed to another fALS mutant TDP-43 (A315T) transgenic strain with the TDP-43 expression cassette integrated into a different chromosome (chromosome II), TDP-43 tg2. We observed significant improvement in locomotion, consistent with the RNAi screening results, in 9 of the 17 loss of function cross strains (Fig 2A and 2B, Table 2).

Improvements in TDP-43 tg *C. elegans* motor dysfunction could be the result of decreased TDP-43 transgene expression, or reduced levels of total TDP-43 protein or phosphorylation at

**Table 2. Loss of function mutations in TDP-43 modifier genes.**

| C. elegans gene | HUGO symbol | Allele tested | Behavior | TDP-43 protein |
|---|---|---|---|---|
| *Energy production and metabolism* | | | | |
| F55G1.5 | SLC25A18 | *gk1250* | NC | NC |
| *paqr-1* | ADIPOR1/2 | *tm3262* | ++ | ++ |
| *Extracellular matrix and cytoskeleton* | | | | |
| *gly-8* | GALNT11 | *tm1156* | ++ | ++ |
| *hse-5* | GLCE | *tm472* | ++ | ++ |
| *sax-2* | FRYL | *ky216* | ++ | ++ |
| *vab-9* | TMEM47 | *gk398617* | ++ | NC |
| *zig-3* | HMCN1 | *tm924* | ++ | ++ |
| *Ion transport* | | | | |
| *unc-77* | NALCN | *e625* | NC | NC |
| *Nucleic acid function* | | | | |
| *tbx-11* | TBR1 | *gk681547* | NC | NC |
| *umps-1* | UMPS | *zu456* | NC | NC |
| *Proteostasis* | | | | |
| *gpx-7* | GPX4 | *tm2166* | NC | NC |
| *pcp-5* | PRCP | *gk446* | ++ | NC |
| *Signaling* | | | | |
| F31E3.2 | SGK494 | *ok2044* | - - | NC |
| F40B5.2 | TMPPE | *gk920502* | NC | NC |

NC = no change; ++ = suppressor; - - = enhancer

amino acid residues S409 and S410 [16]. To determine if TDP-43 mRNA expression levels were altered in these strains, we performed quantitative reverse-transcription PCR (qRT-PCR) and found that expression of the TDP-43 is not decreased in the mutant strains (S1A and S1B Fig). Therefore, we tested whether the suppressor mutants crossed to TDP-43 tg exhibited changes in TDP-43 protein abundance or phosphorylation by immunoblot. We found that 5 of the mutants tested had decreased levels of total or phosphorylated TDP-43 protein (Fig 2C–2G). Interestingly, these results indicate the remaining 4 suppressor genes that improve TDP-43 motor dysfunction do so without a significant impact on TDP-43 phosphorylation or protein levels (S1 Fig).

Based on its predicted function and consistent partial suppression of TDP-43 tg motor phenotypes in *C. elegans*, we chose *hse-5* for more detailed characterization. *hse-5(tm472)* animals have similar locomotion to non-Tg (N2) *C. elegans* (S2A Fig), indicating the improved motility of TDP-43 tg; *hse-5(tm472)* is not due to hyperactivity. *hse-5(tm472)* was initially identified as a suppressor of mutant human TDP-43 (M337V). To see whether it can also suppress wild-type TDP-43 toxicity, we crossed *hse-5(tm472)* with *C. elegans* expressing wild-type human TDP-43 pan-neuronally (WT TDP-43 tg; *hse-5(tm472)*) and assayed their motor function. We found that *hse-5(tm472)* can also suppress motility defects of WT TDP-43 tg (S2B Fig). To explore the possibility that *hse-5* generally protects against aggregating or hyperphosphorylated neurodegenerative disease proteins, we crossed *hse-5(tm472)* with *C. elegans* expressing wild-type or mutant tau protein (tau tg (WT); *hse-5(tm472)* or tau tg (V337M); *hse-5(tm472)*). However, *hse-5(tm472)* did not suppress tau tg motor defects (S2C and S2D Fig), and conversely somewhat worsened tau tg motility. Therefore, *hse-5(tm472)* may be a specific suppressor of TDP-43.

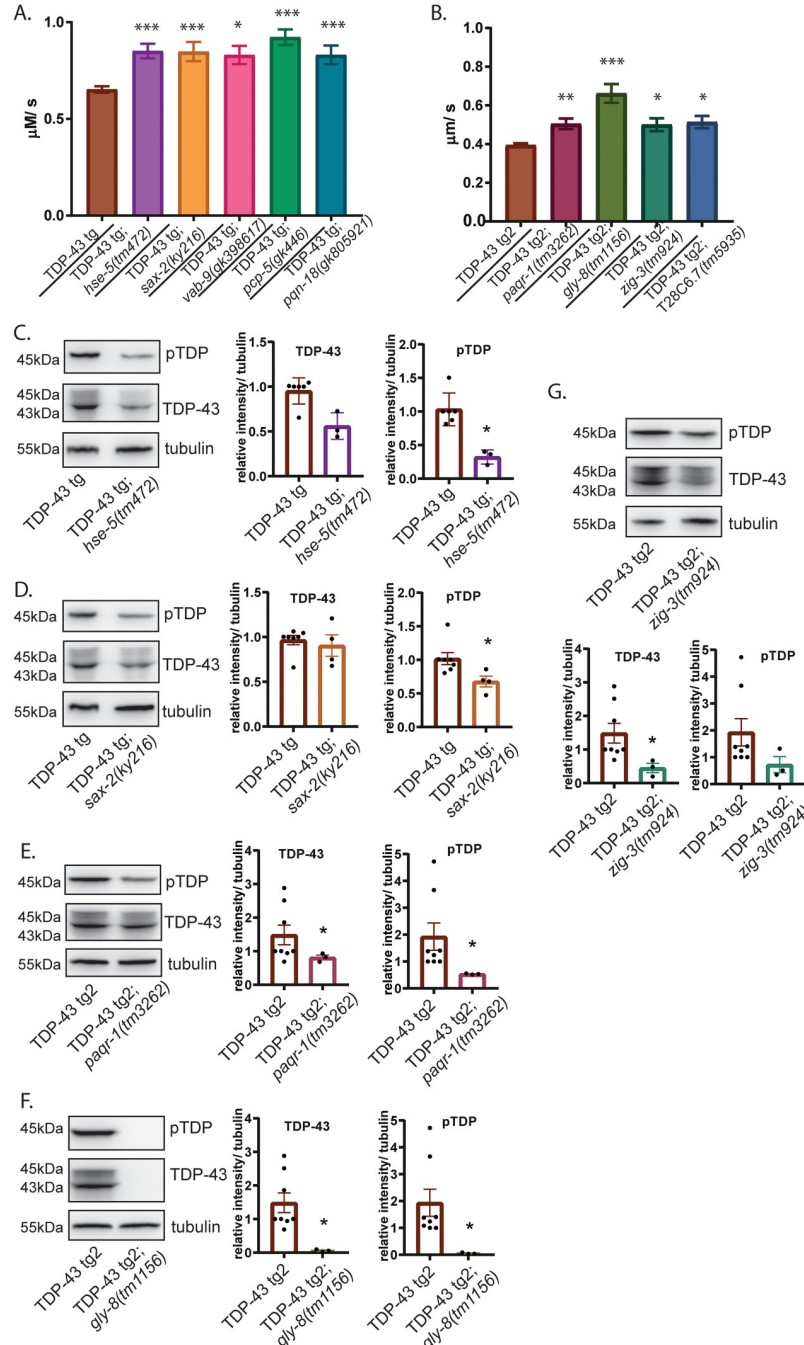

**Fig 2. Suppressors of TDP-43 improve *C. elegans* motor function and reduce levels of toxic TDP-43.** (*A*) Motor function of animals doubly homozygous for the human mutant TDP-43 (M337V) transgene, TDP-43 tg, and loss-of-function mutations for the indicated suppressor genes was assessed. Dispersal velocity of developmentally staged L4 larvae was measured by calculating the radial distance traveled from a designated central starting point over time, N>200 for all strains tested. Significance was evaluated using one-way analysis of variance with Tukey's multiple comparison test among strains tested. \*\*\*p<0.0001, \*p<0.01 versus TDP-43 tg. non-Tg (N2) animals move 3.712 μm/sec. (*B*) To assay the effects of loss-of-function mutations in suppressor genes present on the same chromosome as the TDP-43 tg transgene (Chr IV), a second human mutant TDP-43 (A315T) transgene was utilized on a different chromosome (Chr II), TDP-43 tg2. Animals doubly homozygous for the TDP-43 tg2 transgene and loss-of-function mutations for the TDP-43 suppressor genes indicated were assessed for motor function as above, N>200 for all strains tested. \*\*\*p<0.0001, \*\*p<0.001, \*p<0.01 versus TDP-43 tg2. (*C-G*) Measurement of levels of total and phosphorylated

TDP-43 protein by immunoblot. Data shown are representative of triplicate independent experiments. Graphs plot relative total TDP-43 or phosphorylated TDP-43 signal normalized to tubulin protein levels from three or more independent replicate experiments. Significance was evaluated using Mann-Whitney test between strains tested. *p<0.05.

Although predominantly expressed in the hypodermis and intestine, *hse-5* has been previously shown in *C. elegans* to control axon pathfinding, and loss of *hse-5* function (*hse-5 (tm472)*) causes nerve cord defasciculation, aberrant axonal branching and extra processes [37, 38]. In addition, *hse-5* may regulate the connectivity of neurons controlling stereotyped behaviors such as *C. elegans* male mating [39]. The human homolog of *hse-5*, *GLCE*, encodes the heparan sulfate-modifying enzyme glucuronic acid epimerase (GLCE). As a linear polysaccharide, heparan sulfate (HS) can be covalently attached to a protein core to form proteoglycans. HS proteoglycans play roles in both cellular and extracellular functions: they comprise part of the basement membrane matrix, are present in secretory vesicles, interact with regulatory factors such as cytokines and growth factors, facilitate cell-cell and cell-ECM interactions, and act as receptors for numerous signaling processes [40]. GLCE catalyzes the inversion stereochemistry of D-glucuronic acid to L-iduronic acid, which increases the flexibility of HS and promotes HS mediated ligand interaction and cellular signaling [41]. GLCE is an essential enzyme, and its targeted disruption results in murine neonatal lethality [42]. Additionally, loss of GLCE has been shown to promote neuritogenesis, supporting a neuronal role for this protein [43].

*hse-5(tm472)* may protect against TDP-43 neurotoxicity through a pro-longevity pathway. Therefore, we assayed whether *hse-5(tm472)* protects against the shortened lifespan of TDP-43 tg *C. elegans*. Surprisingly, *hse-5(tm472)* was significantly long-lived relative to non-Tg *C. elegans*, but did not rescue the shortened lifespan of TDP-43 tg (S2E Fig). TDP-43 tg animals display degeneration of specific neuronal types with aging, including gamma-aminobutyric acid (GABA) positive inhibitory motor neurons [16]. *hse-5(tm472)* may directly protect neuron cell viability, resulting in the improved movement observed in Fig 2A. *C. elegans* neuronal integrity can be assessed *in vivo* using a GABAergic D-type motor neuron reporter, P*unc-47*::GFP, which drives expression of GFP in all 19 D-type motor neurons [44]. To determine whether motor neurons are protected in TDP-43 tg; *hse-5(tm472)* animals, we used the P*unc-47*::GFP fluorescent reporter to score intact GABAergic motor neurons *in vivo*. We first scored animals at larval stage 4 (L4), when the first apparent neuronal loss occurs with TDP-43 tg. Surprisingly, we found that TDP-43 tg; *hse-5(tm472)* on average lost significantly more neurons than TDP-43 tg alone (Fig 3A). However, by day 1 of adulthood, there are no significant differences in numbers of neurons lost between TDP-43 tg and TDP-43 tg; *hse-5(tm472)* (Fig 3B). Therefore, the behavioral improvement seen in Fig 2A is not the result of *hse-5(tm472)* protection against aging or neurodegeneration.

Loss of *hse-5* causes aberrant axonal connectivity including non-canonical axonal branching [37]; these abnormalities were apparent when examining *hse-5(tm472)* mutant animals expressing the P*unc-47*::GFP reporter (Fig 3C–3F). In fact, the additional branches and processes appeared to create a more complex neuronal network in *C. elegans*, with non-standard axonal contacts between adjacent neurons and increased nerve cord contacts for some neurons (Fig 3E). When we quantitated the number of axonal abnormalities observed in individual worms, we found that while both *hse-5(tm472)* and TDP-43 tg alone had moderate numbers of axonal abnormalities, the TDP-43 tg; *hse-5(tm472)* had significantly more (Fig 3C). *hse-5 (tm472)* may restore neuronal function in TDP-43 tg animals. One assay to measure synaptic transmission determines the sensitivity of *C. elegans* to the acetycholinesterase inhibitor

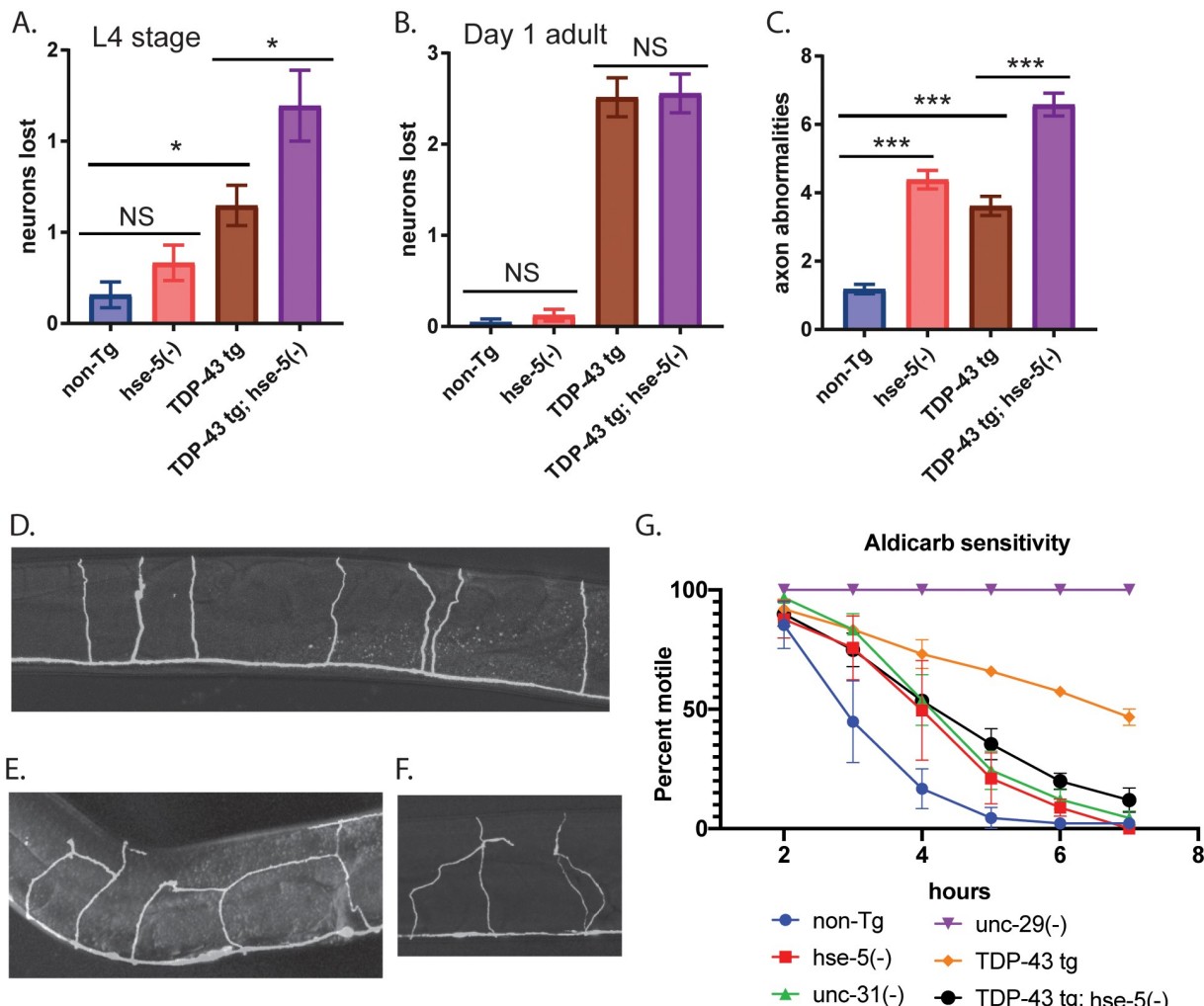

**Fig 3. hse-5 loss of function restores synaptic transmission in TDP-43 tg *C. elegans*.** (*A*) To assess neurodegeneration, GFP-labeled D-type GABAergic motor neurons were counted at L4 stage *in vivo* in living worms. TDP-43 tg; *hse-5(tm472)* animals lose slightly more neurons than TDP-43 tg alone, N>33 for all strains tested. Significance was evaluated using one-way analysis of variance with Tukey's multiple comparison test among strains tested. *p<0.05, non-Tg versus TDP-43 tg and TDP-43 tg versus TDP-43 tg; *hse-5(tm472)*, NS = not significant. (*B*) At day 1 adult, there are no differences in the number of D-type GABAergic motor neurons between TDP-43 tg versus TDP-43 tg; *hse-5(tm472)*, N>33 for all strains tested. NS = not significant. p<0.0001 for non-Tg versus TDP-43tg and TDP-43 tg; *hse-5(tm472)*. (*C*) *hse-5(tm472)* exhibit numerous axonal abnormalities including aberrant branching, looping, and inter-axonal connections. These non-canonical processes are increased in TDP-43 tg; *hse-5(tm472)* animals, N>33 for all strains tested. ***p<0.0001 for non-Tg versus *hse-5 (tm472)*, TDP-43tg, and TDP-43 tg; *hse-5(tm472)*, and for TDP-43tg versus TDP-43 tg; *hse-5(tm472)*. (*D*) Wild-type control axons ascend linearly from the ventral nerve cord without branching, looping, or inter-axonal connections. (*E-F*) TDP-43 tg; *hse-5(tm472)* axons frequently exhibit unusual branches, loops, or inter-axonal connections. *(G)* Pre-synaptic transmission was assessed using an aldicarb sensitivity assay. Animals were scored at the indicated time points for paralysis. Triplicate independent experiments are plotted. TDP-43 tg animals (orange diamond, p<0.005 at hours 3–7) are strongly resistant to aldicarb relative to non-Tg animals (blue circle) indicating decreased or defective synaptic transmission, while *hse-5(tm472)* animals (red square, p<0.05 at hours 3–4) are more modestly resistant to aldicarb. TDP-43 tg; *hse-5(tm472)* animals (black circle, p<0.05 at hours 3–5) have synaptic transmission restored to *hse-5(tm472)* levels. *unc-29(e1072)* (purple inverted triangle, p<0.05 at hours 3–7) and *unc-31(e928)* (green triangle, p<0.05 at hours 3 and 4) are controls for decreased aldicarb sensitivity. Significance was evaluated using two-way ANOVA with Tukey's multiple comparisons test; p-values compared to non-Tg.

aldicarb. Aldicarb prevents acetycholinesterase from breaking down acetycholine in the pre-synaptic cleft. This results in a build-up of acetylcholine, hyperactivation of cholinergic receptors, and eventual paralysis due to hypercontraction. Animals with defects in pre-synaptic transmission can have reduced pre-synaptic acetycholine release, making them resistant to the

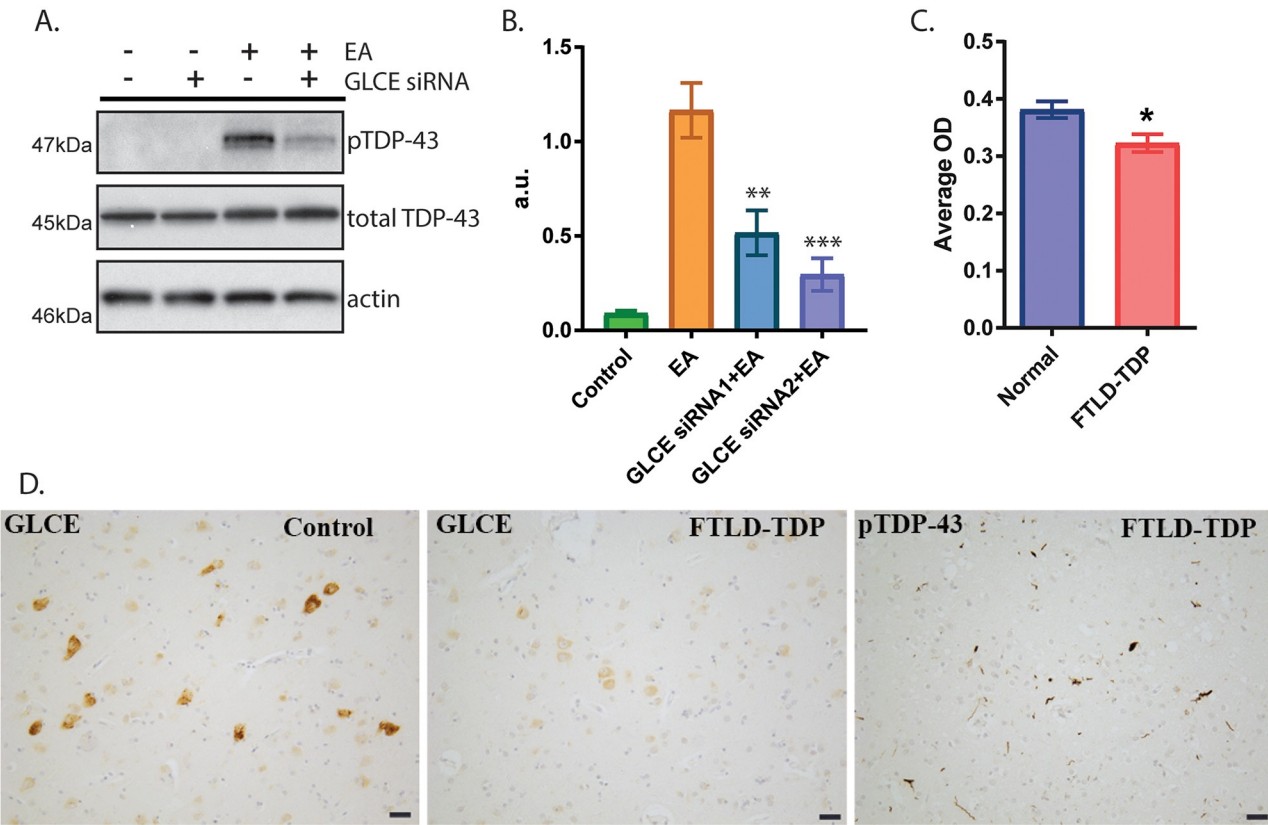

**Fig 4. GLCE changes in mammalian cultured cells and in FTLD-TDP.** (*A-B*) Cultured HEK293 cells treated with ethacrynic acid (EA) accumulate robust levels of phosphorylated TDP-43. (*A*) Representative immunoblots of cells treated with *GLCE*-targeted siRNA exhibit reduced pTDP accumulation in the presence of EA. *(B)* Triplicate independent experiments were quantified and graphed. Significance was evaluated using one-way analysis of variance with Tukey's multiple comparison test among strains tested. **p = 0.0029, ***p = 0.0002 versus EA treated. (*C-D*) GLCE immunoreactivity is reduced in patients with FTLD-TDP compared with normal controls. (*C*) Quantitation of GLCE immunoreactivity in the deep layers of the frontal cortex (mid-frontal gyrus) from 14 patients with FTLD-TDP and 12 normal controls. Significance was evaluated using an unpaired t-test, *p = 0.0116. (*D*) Representative images of GLCE immunostaining in the frontal cortex of patients and controls. pTDP-43 accumulation is apparent in FTLD-TDP patients in this same brain region. Scale bar = 50um.

effects of aldicarb [45]. TDP-43 tg animals exhibit significant resistance to aldicarb, indicating they have a severe defect in synaptic transmission (Fig 3G). *hse-5(tm472)* has a moderate defect in synaptic transmission independent of TDP-43 tg. However, we found that TDP-43 tg; *hse-5 (tm472)* aldicarb sensitivity is similar to the more modest defects of *hse-5(tm472)* alone, indicating that *hse-5(tm472)* restores TDP-43 tg synaptic transmission.

Loss of *C. elegans hse-5* protects against accumulation of phosphorylated TDP-43 (Fig 2C). However, it is unknown whether reduced expression of the human homolog, *GLCE*, would similarly protect against phosphorylated TDP-43 accumulation. To test this, we utilized a mammalian cell culture model that exhibits robust phosphorylation of endogenous TDP-43 driven by an oxidative chemical trigger [46]. Cells were treated with a *GLCE*-targeting siRNA and tested for their accumulation of phosphorylated TDP-43. We found that *GLCE*-targeted siRNA treatment reduced TDP-43 phosphorylation by 60–80% (Fig 4A and 4B), indicating a functional genetic interaction between *hse-5*/ GLCE and TDP-43 remains conserved between *C. elegans* and mammals.

To test whether GLCE protein alterations occur in a human TDP-43 proteinopathy disorder, we examined GLCE immunoreactivity in the frontal cortex of 14 FTLD-TDP patients and

12 normal control cases. We found that GLCE immunoreactivity is robust in the frontal cortex of normal control subjects. In contrast, GLCE immunopositive neurons are sparse in the frontal cortex of FTLD-TDP patients and the neurons that are immunoreactive display much weaker staining. Quantitative immunohistochemistry within the deep layers of the frontal cortex (layers 5/6) revealed that the average optical density of GLCE positive immunostaining is significantly lower in patients with FTLD-TDP compared to controls without TDP-43 proteinopathy (Fig 4C and 4D). All FTLD-TDP patients exhibit robust phosphorylated TDP-43 pathology within this same brain region. These data indicate that decreased GLCE is a feature of FTLD-TDP and may represent a protective response to extend neuron function during disease.

## Discussion

We have conducted an RNAi screen of 16,757 RNAi clones, targeting 86% of the *C. elegans* genome, for suppressors of TDP-43 toxicity in *C. elegans*. We found 46 genes that suppressed TDP-43 motor dysfunction following RNAi-targeted reduction in gene expression, 24 of which had human homologs. None of these genes have been previously implicated as potential modifiers of ALS through genome-wide association studies (GWAS) [47]. These genes fall into a variety of pathways, including energy production and metabolism, extracellular matrix and cytoskeleton, ion transport, nucleic acids, proteostasis, and cell signaling (Table 1). These TDP-43 suppressor genes increase our understanding of the mechanisms of TDP-43 toxicity by nominating involved cellular pathways and processes. Further, these genes and pathways represent potential new therapeutic targets for treating TDP-43 proteinopathies such as FTLD-TDP.

We identified 5 genes that participate in energy production and metabolism: *cox-10*, F23F12.3, F55G1.5, *paqr-1*, and *cox-6A*. Three of these, *cox-10*, *cox-6A*, and F23F12.3 function in mitochondria. Cytochrome C oxidase (COX) is the terminal enzyme in the electron transport chain and *cox-10*/ COX10 and *cox-6A*/ COX6A1 are a COX assembly factor and structural component, respectively. Mutations in COX6A1 and COX10 are associated with Charcot-Marie-Tooth (CMT) disease and mitochondrial disorders [48–50]. F55G1.5/ SLC25A18 catalyzes the unidirectional transport of glutamate into mitochondria [51]. Two of the genes, F23F12.3 and *paqr-1*, are involved in fatty acid metabolism. F23F12.3/ SLC22A5 is a sodium dependent carnitine cotransporter required for beta-oxidation of long-chain fatty acids to produce ATP and is associated with primary systemic carnitine deficiency [52]. *paqr-1*/ ADIPOR1/2 is an adiponectin receptor mediating AMPK and PPARα activities to regulates glucose uptake and fatty acid oxidation by adiponectin [53].

Six of the suppressors are involved in the extracellular matrix and cytoskeleton: *col-89*, *gly-8*, *hse-5*, *sax-2*, *vab-9*, and *zig-3*. *col-89*/ COL22A1 functions as a collagen in cell adhesion ligand maintaining vascular integrity [54]. *gly-8*/ GALNT11 is an N-acetylgalactosaminyl-transferase initiating O-linked glycosylation that regulates left-right asymmetry (through Notch signaling) and cilia during development [55]. *hse-5*/ GLCE is a D-glucuronyl C5-epimerase that modifies heparan sulfate, a key component of the extracellular matrix, and regulates neuronal migration and axonal pathfinding [56]. *sax-2*/ FRYL binds microtubules and aids in the maintenance of nerve ring structure and control of sensory dendrite termination points in *C. elegans* [57–59]. *vab-9*/ TMEM47 is a transmembrane protein localized to the ER and plasma membrane that regulates organization of actin at tight junctions [60]. Finally, *zig-3*/ HMCN1 is a Ca2+ binding extracellular matrix protein that maintains mature axon positioning. Mutations in *HMCN1* are associated with age-related macular degeneration [61].

We found three genes involved in ion transport: *cnnm-3*, C13B4.1, and *unc-77*. *cnnm-3/ CNNM4* is a metal ion transporter, and mutations in *CNNM4* cause Jalili syndrome, a disorder that includes retinal cone-rod dystrophy and amelogenesis imperfecta [62, 63]. C13B4.1/ *FRRS1* is a ferric chelate reductase that reduces ferric to ferrous iron before its transport from the endosome to the cytoplasm. *unc-77/ NALCN* is a voltage-gated sodium channel that propagates neuronal activity from cell bodies to synapses. Recessive mutations in *NALCN* cause infantile hypotonia, developmental delay, and retardation [64].

Three genes have nucleic acid functions: *dna-2*, *tbx-11*, and *umps-1*. *dna-2/ DNA2* is a DNA-dependent ATPase, helicase, and endonuclease that maintains mitochondrial and nuclear DNA stability. Mutations in *DNA2* cause progressive external ophthalmoplegia with mitochondrial DNA deletions-6 and Seckel syndrome [65]. *tbx-11/ TBR1* is a T-box transcription factor that regulates neuronal migration and axon pathfinding [66]. *umps-1/ UMPS* is a uridine monophosphate synthetase that catalyzes pyrimidine biosynthesis by converting orotate to uridine monophosphate. Mutations in *UMPS* are associated with orotic aciduria [67].

We also found three genes involved in proteostasis: C47E12.3, *gpx-7*, and *pcp-5*. C47E12.3/ *EDEM1* is an ER degradation-enhancing alpha-mannosidase that targets misfolded glycoproteins for degradation [68]. *gpx-7/ GPX4* is a glutathione peroxidase that reduces phospholipid hydroperoxides within membranes and lipoproteins, inhibiting lipid peroxidation. Mutations in *GPX4* cause spondylometaphyseal dysplasia [69]. *pcp-5/ PRCP* is a prolylcarboxypeptidase that cleaves proline-linked C-terminal amino acids in the lysosome.

Three genes have roles in cell signaling: F31E3.2, F40B5.2, and Y44E3A.4. F31E3.2/ *SGK494* is a serine/ threonine kinase, F40B5.2/ *TMPPE* is a transmembrane metallophosphoesterase, and Y44EA.4/ *SH3KBP1* is an adaptor protein that regulates diverse signal transduction pathways.

Of the genes identified in our screen, several have previously been implicated in ALS pathobiology by the work of others. For instance, *HMCN1*, the human homolog of *zig-3*, was identified as being one of several cell adhesion and extracellular matrix genes aberrantly spliced in sporadic cases of ALS [70]. In addition, expression of the mouse homolog of *TBR1/ tbx-11* is downregulated in a conditional TDP-43 knockout model [71], and GPX4/ *gpx-7* is differentially expressed in a transgenic mouse model expressing human A315T mutant TDP-43 [72], indicating potentially conserved roles for these genes in disease that warrants further testing. Taken together, the previous identification of some of the hits from our screen reinforces the potential disease relevance of our findings. However, most of the TDP-43 modifying genes identified in our study have not been previously implicated in ALS or other TDP-43 proteinopathies. The *C. elegans* model used here expresses human mutant TDP-43 pan-neuronally, resulting in severe progressive motor impairment, neurodegeneration, and shortened lifespan. Most prominently, these animals accumulate robust phosphorylated TDP-43, making this a powerful model to study the regulation and biological effects of post-translationally modified TDP-43 [16].

17 of the RNAi-mediated suppressors were tested for protection against TDP-43 tg phenotypes using deletion loss-of-function mutants, and 9 of these recapitulated the RNAi results. Of those with human homologs, *paqr-1/ ADIPOR1/2*, *gly-8/ GALNT11*, *hse-5/ GLCE*, *sax-2/ FRYL*, and *zig-3/ HMCN1* suppressed TDP-43-dependent motor dysfunction and reduced accumulation of total or phosphorylated TDP-43. *vab-9/ TMEM47* and *pcp-5/ PRCP* suppressed TDP-43 tg motor dysfunction but did not significantly change accumulation of TDP-43, which indicates these genes may modulate TDP-43 toxicity indirectly rather than through a direct effect on TDP-43 protein.

Our subsequent analysis focused on the *C. elegans* homolog of *GLCE*, *hse-5*, a heparan sulfate epimerase. *hse-5(tm472)* protects against TDP-43-driven motor dysfunction and prevents accumulation of phosphorylated TDP-43. We found that while *hse-5(tm472)* does not prevent TDP-43-driven neurodegeneration, TDP-43 tg; *hse-5(tm472)* animals have numerous aberrant axonal connections among neurons and between nerve cords, consistent with previous work identifying roles for *hse-5* in axonal outgrowth and regeneration [37, 38, 73–75]. We also find that loss of *hse-5* partially restored TDP-43 tg synaptic transmission, identifying a novel role for *hse-5* in synapse function. *hse-5* mediated changes in neuronal connectivity and function may account for the improved movement of TDP-43 tg; *hse-5(tm472)* animals by maintaining neuronal activity better in the face of neuronal loss. This could also be through the developmental role of *hse-5(tm472)* in mediating extracellular matrix cues driving axon guidance along the midline or neuroblast migration [38, 76]. Alternatively, *hse-5(tm472)* may exacerbate TDP-43 neurodegeneration (Fig 3A), but still have a net protective effect by promoting clearance of phosphorylated TDP-43 and delaying neuronal dysfunction. In support of this, we found that *GLCE*-targeted siRNAs in human cultured cells reduced accumulation of pTDP. Furthermore, in FTLD-TDP patient brain neurons, we found decreased GLCE protein compared to controls. Reduction in GLCE in FTLD-TDP may represent a protective response in neurons to decrease accumulation of toxic phosphorylated TDP-43. In fact, the expression of a number of extracellular matrix genes was found to be significantly disrupted in sporadic ALS motor neurons, highlighting the importance of local cell to cell communication between neighboring neurons in parallel, as well as between synapse connected neurons in series, to support motor neuron health [70]. Based on these results, our working model supports two parallel mechanisms by which loss of *hse-5/ GLCE* function is protective. First, normal activity of *hse-5/ GLCE* promotes TDP-43 phosphorylation, likely through indirect effects on cell homeostasis or transmission of intercellular signaling. Second, *hse-5/ GLCE* also controls axonal outgrowth and synaptic transmission, influencing neuronal signaling. Loss of *hse-5/ GLCE* thus simultaneously decreases accumulation of phosphorylated TDP-43 and alters synaptic transmission, protecting against TDP-43-induced neuronal dysfunction.

In summary, this work has identified new target genes and molecular pathways controlling TDP-43 toxicity. Additional investigation of these genes in mammalian systems will be important future work. Preliminary characterization of GLCE suggests it can act to detoxify pTDP and is a therapeutic target worth further examination.

## Materials and methods

### Ethics statement

We obtained samples of postmortem tissue from the University of Washington Alzheimer's Disease Research Center (ADRC) Neuropathology Core (PI, Dr. C. Dirk Keene) after receiving human subjects approval (University of Washington Human Subjects Division approval: HSD# 06-0492-E/A 01). FTLD-TDP cases were selected on the basis of having an autopsy-confirmed neuropathological diagnosis of FTLD with TDP-43 deposits according to consensus criteria [81]. Control samples were from neurologically healthy control participants, who were of a similar age with low levels of AD pathology (Braak stage III or less and CERAD scores of none or sparse) [82, 83]. All patients or their next of kin provided written, informed consent.

### *C. elegans* strains and transgenics

Wild-type *C. elegans* (Bristol strain N2) was maintained as previously described [77]. Transgenic strains used were CK674 *eri-1(mg366)*; *lin-15b(n744)*; *bkIs674*[P*snb-1*::hTDP-43 (M337V)+P*myo-2*::dsRED] (primary RNAi screen and retesting), CK423 *bkIs423*[P*snb-1*::

hTDP-43(M337V)+P*myo-2*::dsRED], and CK426 *bkIs426*[P*snb-1*::hTDP-43(A315T)+P*myo-2*:: dsRED] [16, 78]. GR1373 *eri-1(mg366)* was used for screen counter-selection. CK423 was crossed with the following loss-of-function alleles: *vab-9*(*gk398617*) II, T24H10.4(*gk637193*) II, *hse-5*(*tm472*) III, *sax-2*(*ky216*) III, *tbx-11*(*gk681547*) III, F31E3.2(*ok2044*) III, *pcp-5*(*gk446*) III, *umps-1*(*zu456*) III, F40B5.2(*gk920502*) X, and *pqn-18*(*gk805921*) X. CK426 was crossed with the following loss-of-function alleles: *gly-8*(*tm1156*) III, *unc-77*(*e625*) IV, *paqr-1*(*tm3262*) IV, F55G1.5(*gk1250*) IV, T28C6.7(*tm5935*) IV, C47E12.3(*ok2898*) IV, *gpx-7*(*tm2166*) X, and *zig-3* (*tm924*) X. *hse-5*(*tm472*) was crossed with CK410 *bkIs410*[P*snb-1*::hTDP-43(WT)+P*myo-2*:: dsRED], CK10 *bkIs10*[P*aex-3*::hTau(V337M)+P*myo-2*::GFP], and CK144 *bkIs144*[P*aex-3*:: hTau(WT)+P*myo-2*::GFP] for motility assays. OH1487 *hse-5(tm472)*, DA509 *unc-31(e928)*, and CB1072 *unc-29(e1072)* were used as controls.

### *C. elegans* RNAi screening

Initial screening of the feeding RNAi Ahringer library was conducted essentially as described [79]. Bacterial cultures were grown over night and 40μL were seeded on twelve well plates of Nematode Growth Medium (NGM) supplemented with 25μg/mL carbenicillin and 2mM iso-propyl-b-D-thiolgalactophyranoside (IPTG) and allowed to dry for two days. Embryos were isolated by hypochlorite treatment of gravid adult worms, placed onto 12 well RNAi plates at a density of approximately 15 eggs per well, and allowed to grow for 10–12 days at 16˚C until the F2 generation were young adults. Controls including RNAi vector alone, RNAi targeting human TDP-43 (suppressor control), and RNAi targeting *unc-22* (enhancer control) were screened in parallel for all experiments. Follow-up screening of putative positives was carried out on individual 60mm plates and were given a semi-quantitative overall score from 0 to 5, where 0 is unable to differentiate from the empty L4440 vector control, and 5 is complete suppression to N2 wild-type behavior. All RNAi clones from the Ahringer library that passed initial screening were analyzed by Sanger sequencing to validate clone targeting.

### Behavioral analysis

Assessments of *C. elegans* locomotion were carried out as previously described with minor modifications [16]. In brief, 15–20 animals were placed at the center of a 100mm plate of 5x concentrated peptone nematode growth media uniformly seeded with OP-50 bacteria. Animals were allowed to move freely for 30–60 minutes, and the radial distance traveled from the start point was recorded. Distance traveled was converted to micrometers per minute. Statistical significance was analyzed using Prism statistical software.

### Neurodegeneration and axon abnormality assays

Strains with GFP marked gamma-aminobutyric acid (GABA)-ergic motor neurons were generated by crossing *hse-5(tm472)* with CK423 and the reporter strain EG1285 *oxIs12*[P*unc-47*:: GFP + *lin-15(+)*]. Animals were grown to the indicated stage at 20˚C before scoring. Living animals were immobilized on a 2% agarose pad with 0.01% sodium azide, and intact GABA-ergic neurons (L4 or day 1 adult) or axon abnormalities (L4 stage) were scored under fluorescence microscopy on a DeltaVision Elite (GE) imaging system using an Olympus 60x oil objective. Statistical significance was analyzed using Prism statistical software.

### Microscopy, image acquisition, and processing

Images were acquired using a Leica SP5 confocal microscope with a 40x oil immersion lens. Z-plane stacked images were flattened into a maximum intensity projection using Imaris x64

software. Images were minimally processed using Adobe Photoshop to enhance visibility of fluorescent signal.

### Aldicarb sensitivity assay

*C. elegans* were grown to L4 stage at 20˚C, and then transferred to NGM plates containing the indicated concentrations of aldicarb (Sigma). Animals were scored for paralysis using a gentle touch on the head and tail with a platinum wire [45].

### Immunoblotting

Stage-matched day 1 adult *C. elegans* were harvested and snap frozen. Protein was extracted by resuspending pellets in 1X sample buffer, 20 s sonication, and 5 minutes boiling. 10 μL of samples were loaded onto precast 4–15% gradient Tris: HCL gels (Biorad). Total human TDP-43 was detected with mouse monoclonal antibody ab57105 (Abcam) and phosphorylated TDP-43 was detected with mouse monoclonal antibody TIP-PTD-M01 (CosmoBio). Beta-Tubulin antibody E7 (DSHB) was used as a load control for all samples.

### Quantitative reverse-transcription PCR (qRT-PCR)

RNA was extracted using Trizol (Sigma), and cDNA was prepared using iScript Reverse Transcription Supermix for RT-qPCR (Bio-Rad). Each genotype was tested with three biological replicates. qPCR was performed using the iTaq Universal SYBR Green Supermix kit (Bio-Rad) on a CFX Connect Real-Time PCR Instrument (BioRad). Data were normalized within samples using an internal reference control gene (*act-1*).

### Lifespan assay

*C. elegans* were grown at 20˚C following a short (4–6 hour) egglay to L4 stage on NGM plates seeded with *E. coli* OP50, and then transferred onto seeded NGM plates with added 5-fluoro-2'-deoxyuridine (FUDR, 0.05 mg/mL) to inhibit growth of progeny. Individuals were scored every 2–3 days by gentle touching with a platinum wire. Failure to respond to touch was scored as dead. Statistical analysis was performed using GraphPad Prism software.

### Cell culture, RNA interference, and EA treatment

HEK293 cells were cultured under standard tissue culture conditions (DMEM, 10% defined fetal bovine serum, Penicillin (1000 IU/mL) Streptomycin (1000μg/mL)) as previously described [80]. RNA interference was conducted following manufacture's protocol (RNAi-MAX, Invitrogen). Induction of pTDP-43 with ethacrynic acid was conducted as previously described [80].

### Immunohistochemistry and quantification

Immunohistochemistry was performed on paraffin embedded frontal cortex sections from 14 patients with FTLD-TDP and 12 normal control cases. Antigen retrieval consisted of autoclaving in a citrate buffer. Sections were treated for endogenous peroxidases, blocked in 5% non-fat milk in PBS, and incubated overnight at 4˚C with anti-GLCE polyclonal antibody (ThermoFisher Sci, 1:75). Biotinylated secondary antibody was applied, followed by incubation in an avidin-biotin complex (Vector's Vectastain Elite ABC kit, Burlingame, CA) and the reaction product was visualized with 0.05% diaminobenzidine (DAB)/0.01% hydrogen peroxide in PBS. FTLD-TDP cases were also stained with an anti-phospho TDP-43 antibody (409/410 CosmoBio, 1:1000) to confirm TDP-43 pathology. In order to minimize variability, sections

from all cases (normal and affected subjects) were stained simultaneously. Digital images were obtained using a Leica DM6 microscope with the DFC 7000 digital camera and LAS X imaging software. HALO digital image software (Indica Labs) was used to quantify GLCE immunoreactivity in the deep layers of the frontal cortex (layers 5/6) using the "Area Quantification" setting. Data represent the average optical density value of DAB staining within the region of interest and are displayed as mean +/- SEM. A two tailed Student's t-test was used to assess differences in GLCE expression between cases and controls.

## Supporting information

**S1 Fig. Characterization of TDP-43 transgene expression levels and immunoblot analysis of suppressors without an effect on TDP-43 protein.** *(A-B)* Suppressors of TDP-43 motor function do not have decreased transgene expression levels. *(A)* Expression of the TDP-43 transgene was assessed in triplicate independent replicate samples using quantitative reverse transcription PCR (qRT-PCR), and signal normalized to expression of an internal reference gene *act-1*. $p > 0.05$ for all samples tested versus TDP-43 tg. *(B)* To assay the effects of loss-of-function mutations in suppressor genes present on the same chromosome as the TDP-43 tg transgene (Chr IV), a second TDP-43 transgene was utilized on a different chromosome (Chr II), TDP-43 tg2. *p = 0.0371 versus TDP-43 tg2. *(C-F)* Measurement of levels of total and phosphorylated TDP-43 protein by immunoblot. Immunoblot data shown are representative of at least triplicate independent experiments, and graphs plot relative total TDP-43 or phosphorylated TDP-43 signal normalized to tubulin protein levels from three or more independent replicate experiments. *(C)* TDP-43 tg; *vab-9(-)*, *(D)* TDP-43 tg; *pcp-5(-)*, *(E)* TDP-43 tg; *pqn-18(-)*, *(F)* TDP-43 tg2; T28C6.7(-). Significance was evaluated using Mann-Whitney test between strains tested. $p > 0.05$ versus TDP-43 tg for all comparisons.
(TIF)

**S2 Fig. Characterization of *hse-5* in non-Tg, WT TDP-43 tg, and tau tg *C. elegans*.** *(A)* *hse-5(tm472)* do not have significant differences in motility relative to non-transgenic (non-Tg) *C. elegans*, $N > 200$ per genotype. *(B)* Motility defects of *C. elegans* expressing wild-type TDP-43 (WT TDP-43 tg) are significantly improved by *hse-5(tm472)*. $p < 0.0001$, unpaired t-test, $N > 200$ per genotype. *(C)* *hse-5(tm472)* does not significantly alter the motility defects in *C. elegans* expressing wild-type human tau, although it is trending towards worsened motility. $p = 0.052$, unpaired t-test, $N > 130$ per genotype. *(D)* *hse-5(tm472)* significantly worsens motility defects in *C. elegans* expressing mutant V337M human tau. $p = 0.0097$, unpaired t-test, $N > 150$ per genotype. *(E)* *hse-5(tm472)* are significantly long-lived relative to non-Tg animals (N2), $p < 0.0001$, Log-rank test. TDP-43 tg are short-lived relative to non-Tg animals, $p < 0.0001$. However, the lifespan of TDP-43 tg; *hse-5(tm472)* are not significantly different from TDP-43 tg animals alone, $p = 0.143$. *(F)* mRNA expression of *hse-5* does not change in TDP-43 tg relative to non-Tg animals. Two different primer pairs were tested using quantitative reverse transcription real-time PCR (qRT-PCR) on three independent replicate samples. Signal was normalized to the expression an internal reference gene, *act-1*.
(TIF)

## Acknowledgments

We thank Heather Currey, Elaine Loomis, Kim Howard, Josh Hincks, Kelsey Price, Kaili Chickering, Aristide Black, and Ashley Yeung for essential technical assistance. We thank Allison Beller for outstanding administrative support. We thank the National Bioresource Project

(Japan) and *C. elegans* Genetics Center (CGC) for providing strains. We thank WormBase for essential *C. elegans* model organism information. We thank the Developmental Studies Hybridoma Bank (NICHD) for the β-tubulin primary antibody E7.

## Author Contributions

**Conceptualization:** Thomas D. Bird, Brian C. Kraemer.

**Data curation:** Nicole F. Liachko, Aleen D. Saxton, Brian C. Kraemer.

**Formal analysis:** Nicole F. Liachko, Aleen D. Saxton, Pamela J. McMillan, Timothy J. Strovas.

**Funding acquisition:** Nicole F. Liachko, C. Dirk Keene, Brian C. Kraemer.

**Investigation:** Nicole F. Liachko, Aleen D. Saxton, Pamela J. McMillan, Timothy J. Strovas, Brian C. Kraemer.

**Methodology:** Nicole F. Liachko, Aleen D. Saxton, Pamela J. McMillan, Timothy J. Strovas.

**Project administration:** Brian C. Kraemer.

**Resources:** C. Dirk Keene, Thomas D. Bird, Brian C. Kraemer.

**Supervision:** Nicole F. Liachko, Brian C. Kraemer.

**Validation:** Nicole F. Liachko, Aleen D. Saxton, Pamela J. McMillan.

**Visualization:** Nicole F. Liachko, Brian C. Kraemer.

**Writing – original draft:** Nicole F. Liachko, Aleen D. Saxton, Brian C. Kraemer.

**Writing – review & editing:** Nicole F. Liachko, Aleen D. Saxton, Pamela J. McMillan, Timothy J. Strovas, C. Dirk Keene, Thomas D. Bird, Brian C. Kraemer.

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
