## [Decision Letter · Decision Letter 0]

10 Jul 2019

Dear Dr Kraemer,

Thank you very much for submitting your Research Article entitled 'Genome wide analysis reveals heparan sulfate epimerase modulates TDP-43 proteinopathy' to PLOS Genetics. Your manuscript was fully evaluated at the editorial level and by independent peer reviewers. The reviewers appreciated the importance of your screen and the analysis provided, but they raised some substantial concerns about the current manuscript. Based on the reviews, we will not be able to accept this version of the manuscript, but we would be quite willing to review again a revised version that addresses the issues raised by the reviewers. We cannot, of course, promise publication at that time.

Should you decide to revise the manuscript for further consideration here, your revisions should address each of the specific points made by each reviewer. As noted in the comments, some of these can be dealt with by revising manuscript text or including information requested. We will require a detailed list of your responses to the review comments and a description of the changes you have made in the manuscript.

If you decide to revise the manuscript for further consideration at PLOS Genetics, please aim to resubmit within the next 60 days, unless it will take extra time to address the concerns of the reviewers, in which case we would appreciate an expected resubmission date by email to plosgenetics@plos.org.

[LINK]

We are sorry that we cannot be more positive about your manuscript at this stage. Please do not hesitate to contact us if you have any concerns or questions.

Yours sincerely,

Anne C. Hart

Associate Editor

PLOS Genetics

Gregory Barsh

Editor-in-Chief

PLOS Genetics

Reviewer's Responses to Questions

**Comments to the Authors:**

Reviewer #1: This study by Liachko et al perform the first genome-wide RNAi screen for modifiers of TDP-43 M337V toxicity. They find a number of candidates in interesting GO categories such as energy metabolism, protein homeostasis, and ECM/cytoskeleton. They then follow up on a single candidate gene, hse-5. They find that an hse-5 mutant decreases TDP-43 neurodegeneration of GABAergic cells in C. elegans, and that knock down of the ortholog GLCE in a mammaliam model of pTDP-43 is also protective. Finally they see decreased levels of GLCE in human brains with TDP-43 pathology. Overall, this is a well-done, comprehensive screen for modifiers of TDP-43 toxicity. The candidate gene hse-5/GLCE is quite intriguing with solid follow up in cell culture and human neuropath. A few minor additions would strengthen this paper but overall an important, well-done study.

1) For the Western blots of TDP-43 and pTDP-43, some quantification of the results would be helpful. The suppressor hits that did not change TDP-43 and pTDP-43 levels would also be interesting to see.

2) In regards to TDP-43 levels, the authors are careful to avoid saying that the decreased levels are due to improved clearance, but nonetheless it would be important to see if the candidates change TDP-43 mRNA levels. If they do so, it would not disqualify the candidates, but one would have a better, different idea of mechanism of protection.

3) In regards to hse-5, one wonders if adult only (or from L4 onwards) RNAi has an effect and would avoid the potentially protective/potentially harmful axonal abnormalities

4) The screen was performed in a mutant TDP-43 line, however the cell culture and FTLD-TDP would presumably express wild-type TDP-43. Therefore, it would be interesting to see if the hse-5 RNAi or mutant is protective against the WT TDP-43.

5) Also, Table 2 is referred to in the text, but was not included in the manuscript for review.

Reviewer #2: Genome wide analysis reveals heparan sulfate epimerase modulates TDP-43 proteinopathy

The manuscript by Liachko and colleagues is an investigation into finding genetic modifiers of TDP-43 toxicity. These genetic modifiers may inform into the mechanisms underlying TDP-43 linked pathologies, and they may also serve as new targets for therapeutic intervention. The work is done with a well characterized C. elegans model of TDP-43 neuronal toxicity and a genome wide reverse genetics screen using RNA interference to screen for suppression of TDP-43 induced motility phenotypes.

The work is well done, the manuscript is clearly written and logical in its presentation. Furthermore, I would like to commend the authors on completing such a labour intensive genetic screen. The results are novel and likely to aid further research in the field. However, there are a few outstanding issues that need to be addressed before considering the work for publication.

Major points

Unbiased genetic screens are powerful ways to advance understanding. However, sometimes it is not evident by which mechanism a modifier achieves its phenotypic effects. Here the authors sequentially limit their candidate genes to a select few before settling on hse-5/GLCE.

There is no doubt that this gene has an effect on phosopho TDP-43 phenotypes. But the mechanism for the suppression is not sufficiently accounted for.

p. 10 “Preliminary characterization of GLCE suggests it can act to detoxify pTDP and is a therapeutic target worth further examination.”

The choice of focusing on one gene, hse-5 is reasonable, but I have several questions.

How does a potential ECM protein affect the levels of TDP-43 phosphorylation in worms and tissues? What is the working model? The gap in understanding should be addressed, at least as speculation in the discussion.

Has GLCE showed up as a modifier of ALS in GWAS studies?

Where is hse-5 expressed in C. elegans and how does this relate to the expression of mutant TDP-43?

Were both deletion alleles of hse-5 tested?

The fact that hse-5 does not suppress neurodegeneration is unfortunate, perhaps this is the key mechanism, but at the same time maybe it is non-specific. Does hse-5 result in alternative neuronal connections that simply compensate for impaired movement in TDP-43 animals? Do hse-5 animals have generally improved motility? How do they respond to aldicarb and/or levamisole? They say that RNAi candidates working in non-transgenic worms were excluded as part of the selection process, but it is possible that retesting of hse-5 RNAi in N2 worms would miss subtle effects. Thus, do hse-5 mutants have improved motility compared to N2 worms?

Furthermore, a control that is missing is tests versus wild type TDP-43 transgenics. The 2010 Liachko et al. manuscript describes the WT control strains, thus these are likely the floor for the amount of suppression possible with this approach and hse-5 should be tested in these WT TDP-43 strains.

Suppression of phenotypes associated with human disease proteins is of wide interest. Thus, it would be good to know if suppression of TDP-43 toxicity by hse-5 is specific. Does hse-5 protect against other forms of toxicity, perhaps Tau, given the role of phosphorylation in tau toxicity.

Does hse-5 affect the expression of the TDP-43 transgenes? This would be good to know as it could change interpretation of the potential mechanism for reduced pTDP-43. Perhaps for the other 4 candidate genes as well.

Finally, does hse-5 affect lifespan? Also good to know in terms of modifiers of age-dependent phenotypes.

Minor issues

Table 2 is missing.

p. 5 “We found that 5 of the mutants tested had decreased levels of TDP-43 protein accompanied by reduced phosphorylation (Fig. 2C-G). Interestingly, these results indicate the remaining 4 suppressor genes that improve TDP-43 motor dysfunction do so without a direct impact on TDP-43 phosphorylation or protein levels.”

It looks like there is more of a mix of reduced total TDP-43 versus pTDP-43. It is clear that something is happening, but perhaps a simple relative quantification would aid the figure. Also, what happened to TDP-43 in the gly-8 mutants?

Reviewer #3: The authors present data on an unbiased genome-wide RNAi screen for suppressors of tdp-43(M337V) toxicity. After screening 16757 individual gene knockdowns, they identified 46 candidate genes that were required for tdp-43(M337V) toxicity. Based on sequence homology, 24/46 genes appeared to have human homologs. These 24 genes fell into several functional classes. The authors obtained loss-of-function mutants for 17 genes and found that 9 mutants mimicked the RNAi results and suppressed tdp-43(M337V) toxicity. For five of these mutants, the authors showed that overall tdp-43 protein levels were reduced. The authors focus their attention on one gene, hse-5, which consistently suppressed tdp-43(M337V) toxicity. hse-5 appears homologous to the human gene GLCE, which modifies heparin sulfate proteoglycans. An hse-5 mutant, tm472, exhibits improved motor function in the tdp-43(M337V) background. Surprisingly, hse-5(tm472) significantly increases the loss of GABA motor neurons in the tdp-43(M337V) background. The remaining neurons exhibit excessive branching and axonal complexities. This increase in axon abnormalities was additive with tdp-43(M337V). The authors also show that siRNA inhibition of GLCE in human HEK cells substantially reduced endogenous tdp-43 phosphorylation. Finally, the authors examine frontal cortex brain sections from post-mortem FTLD patients and observed significantly reduced GLCE staining in the FTLD patients.

The manuscript is concise and clearly written. The introduction clearly frames the significance of the work. However, the discussion section essentially restates the results and does not offer substantial insights into any of the hits or their potential significance to the field. It would be extremely useful to discuss whether any of these genes or their homologs have been identified in the other screens mentioned by the authors. If so, what can be learned? If not, what might be unique about this tdp-43 model that facilitated their identification? Could the over-expression of mutant human tdp-43 in the background of an endogenous WT worm tdp-43 influence the results of the screen? How might this be addressed? I would also like to see more discussion of why a gene involved in (mostly) extracellular protein modification that is predicting to be localized within the secretory pathway is required for toxicity of a nucleo-cytoplasmic protein. This is not clear and is one of the most interesting conclusions from this work. There is also a lot known about the role of hse-5 in neuronal migration and axon regeneration from previous work, which is basically not discussed. Could these functions possibly explain how hse-5 might be protecting against tdp-43? Additional discussion in this area is warranted.

I also have several major and minor concerns regarding the data as presented:

Major

1. In the screen flow chart, the authors indicate that gene knockdowns that are “different for TDP-43 tg versus non-transgenic” are discarded. Given that the phenotype being screened for is improved motility in the tdp-43(M337V) background, how would one see improved motility in a non-tg (ie wild type) background? Were any hits eliminated based on this criteria? Or was the filtering for gene knockdowns that had similar motility in WT and tdp-43(M337V)? This description needs to be clarified. Along these lines, it would be useful to see the motility of WT in Fig 2A and 2B.

2. Figure 2 C-G, the authors show that 5 loss of function mutants, including hse-5, exhibit reduced phosphorylated tdp-43(M337V) protein levels, as well as total tdp-43(M337V) levels (although paqr mutants look to have similar total tdp but reduced phospho but the authors do not discuss this). The reduction in phospho-tdp could be due to several less interesting effects that have nothing to do with tdp-43, including transgene suppression (ie gly-8 mutants appear to have NO tdp expression) or reduced snb-1 promoter activity and subsequent reductions in tdp mRNA. Were any of these possibilities examined? This is important not only for the authors to interpret their data but also to prevent others from chasing non-existent tdp-43 toxicity mechanisms.

3. The authors state that the results from Fig 2C-G “indicate the remaining 4 suppressor genes improve TDP-43 motor dysfunction without a direct impact on TDP-43 phosphorylation or protein levels.” Given that no data were presented for these 4 mutants, there are no data in the manuscript to support this conclusion.

4. On pg7, para2, the authors state “Loss of C. elegans hse-5 protects against accumulation of phosphorylated TDP-43(Fig. 2C). This is not supported by the data, which show a qualitatively similar decrease in both total tdp-43 protein levels and phospho-tdp-43 levels in hse-5 mutants. Therefore, when hse-5 is lost, there is LESS tdp-43 to protect against.

Minor

1. Why does 2A show motility data in the M337V while 2B shows it in the A315T? Have any suppressors been identified that exhibit allele-specific suppression? Is there something preventing comparisons in the same tdp-43(M337V) background, given that authors went to the trouble to make two unlinked integrated strains?

2. There are two Figure 4C panels

3. The PDF file that I reviewed lacked Table 2, which is referenced in the paper at least twice. Not sure if this was an author or a journal issue.

4. The reduction in GLCE protein in the frontal cortex of FTLD patients is interesting. The hypothesis that reduced GLCE expression in the FTLD patients is protective is quite provocative. If this is true, one might expect GLCE expression levels to be normal in other brain regions not undergoing neurodegeneration. Have other non-degenerating regions been examined? Perhaps this could be more easily addressed in worms. For example, is hse-5 expression down-regulated in tdp-43(M337V) worms?

**Have all data underlying the figures and results presented in the manuscript been provided?**

Reviewer #1: No: Table 2 is referred to in the text, but I cannot find it in the manuscript provided.

Reviewer #2: No: Table 2 was missing.

Reviewer #3: No: numerical data underlying graphs was not provided in a spreadsheet.

PLOS authors have the option to publish the peer review history of their article (what does this mean?). If published, this will include your full peer review and any attached files.

Reviewer #1: No

Reviewer #2: No

Reviewer #3: No

---

## [Decision Letter · Decision Letter 1]

23 Oct 2019

Dear Dr Kraemer,

Thank you very much for submitting your Research Article entitled 'Genome wide analysis reveals heparan sulfate epimerase modulates TDP-43 proteinopathy' to PLOS Genetics. Your manuscript was fully evaluated at the editorial level and by independent peer reviewers. The manuscript describes important work and the revised manuscript has dealt with all substantive concerns raised by the reviewers. There are still some minor concerns raised by reviewer 3. In my opinion as academic editor, these last small concerns do not preclude accepting the manuscript-- but I want to give you the opportunity to read these comments and consider changing the manuscript text. Adopting these reviewer suggested changes may increase clarity and readability of the manuscript. Therefore, I am returning the manuscript to you for "minor revisions". When you resubmit, I will reread the revised manuscript and your response to the reviewer; I do not plan to send the manuscript out to the reviewers again.

[LINK]

Yours sincerely,

Anne C. Hart

Associate Editor

PLOS Genetics

Gregory Barsh

Editor-in-Chief

PLOS Genetics

Reviewer's Responses to Questions

**Comments to the Authors:**

Reviewer #1: The authors have addressed my concerns as stated in the earlier review.

Reviewer #2: This is a much improved manuscript, and the authors have adequately addressed most of my concerns.

Reviewer #3: 1. Were RNAi clones sequenced to confirm their identity? Neither the methods nor the results state that they were sequence verified. Table 2 suggests that many ‘hits’ were not phenocopied using LOF mutants. One possibility for this is that the RNAi clone targeted a different gene. From many RNAi screens, we find that ~10% of Ahringer library RNAi clones do not correspond to the predicted gene when sequenced.

2. There are still some nomenclature issues

a. Allele names in Table 2 are not italicized

b. Mutant name in Fig 2A & B, 3A,B,C,G, S1A,B,S2A-E are not italicized. Also, the allele should be listed, not (-). There is no evidence presented that these alleles are null mutants, as implied by the (-) designation

3. P7 ‘ To confirm mRNA expression of the TDP-43 transgene in these strains…’. I believe the authors meant to say ‘ To determine if TDP43 mRNA expression levels were altered in these strains…’. To me, statements that aim to confirm imply that the authors are trying to prove their hypothesis is true, rather than attempting to disprove their hypothesis.

4. The extended paragraph on hse-5 expression, mutant phenotypes, physiological roles, etc (p7-8) doesn’t seem to belong in the results section and should be moved to the discussion.

5. The authors refer to an interaction between hse-5/GLCE and TDP-43 (p9). Since the authors have not demonstrated that there is a physical interaction between these two proteins, they need to be more precise with this statement and indicate that they are referring to a genetic interaction.

6. I find the first two pages of the discussion section to be largely uninformative. It simply restates the molecular associations for many of the hits from this screen. Virtually all of that information is found in Table 1 and the discussion does not really expand on anything. On the other hand, the discussion that occurs after these sections is much improved.

**Have all data underlying the figures and results presented in the manuscript been provided?**

Reviewer #1: Yes

Reviewer #2: Yes

Reviewer #3: Yes

PLOS authors have the option to publish the peer review history of their article (what does this mean?). If published, this will include your full peer review and any attached files.

Reviewer #1: No

Reviewer #2: No

Reviewer #3: No

---

## [Editor Report · Decision Letter 2]

15 Nov 2019

Dear Dr Kraemer,

We are pleased to inform you that your manuscript entitled "Genome wide analysis reveals heparan sulfate epimerase modulates TDP-43 proteinopathy" has been editorially accepted for publication in PLOS Genetics. Congratulations!

Yours sincerely,

Anne C. Hart

Associate Editor

PLOS Genetics

Gregory Barsh

Editor-in-Chief

PLOS Genetics

Comments from the reviewers (if applicable):

**Data Deposition**

http://datadryad.org/submit?journalID=pgenetics&manu=PGENETICS-D-19-00919R2

**Press Queries**

---

## [Editor Report · Acceptance letter]

5 Dec 2019

PGENETICS-D-19-00919R2 

Genome wide analysis reveals heparan sulfate epimerase modulates TDP-43 proteinopathy 

Dear Dr Kraemer, 

We are pleased to inform you that your manuscript entitled "Genome wide analysis reveals heparan sulfate epimerase modulates TDP-43 proteinopathy" has been formally accepted for publication in PLOS Genetics! Your manuscript is now with our production department and you will be notified of the publication date in due course.

With kind regards,

Nicholas White

PLOS Genetics

On behalf of:
